# Patterns of Blindness in the Navajo Nation: A 9-Year Study

Ryan T. Wallace [1], Michael Murri [1], Lori McCoy [1], Esteban Peralta [2], Jeff H. Pettey [1] and Craig J. Chaya [1,*]

[1] Department of Ophthalmology, John A. Moran Eye Center, University of Utah, Salt Lake City, UT 84112, USA; ryan.wallace@utah.edu (R.T.W.); mike.murri@hsc.utah.edu (M.M.); lori.mccoy@utah.edu (L.M.); jeff.pettey@hsc.utah.edu (J.H.P.)

[2] School of Medicine, University of Utah, Salt Lake City, UT 84112, USA; esteban.peralta@hsc.utah.edu

* Correspondence: craig.chaya@hsc.utah.edu; Tel.: +1-801-213-4125

**Abstract:** The Navajo Nation is the largest Native American reservation by area and citizenship. The study sought to provide the first large-scale examination of ocular pathology within this population. A retrospective review of all Navajo patients seen at Moran Eye Center Navajo Nation Outreach Clinics from 2013 to 2021 for demographics, visual acuity, refractive, and eye pressure data was undergone. Further variables included comorbidity and eye diagnoses among patients at these clinics. Results: First-time patient visits totaled 2251 from 2013 to 2021. The median age was 53 (range, 18 to 92), and clinics had a predominance of female patients (1387:864). Among patients presenting without glasses, 20.67% (198/958), 9.71% (93/958), and 3.13% (30/958) had mild visual impairment (VI), moderate to severe VI, and blindness, respectively. Cataracts were the most common cause of blindness in these patients (40%, 12/30) and the need for glasses was the second most common cause (33%, 10/30). From 2016 to 2021, 17.71% (48/271) of diabetic patients were diagnosed with diabetic retinopathy (DR). Within the subset of Navajo patients that presented without any correction, 73% of bilateral blindness was preventable via glasses prescription or cataract surgery. This study comments on questions of equitable care for Navajo patients.

**Keywords:** Navajo; cataract; eye; equity; native



## 1. Introduction

The Navajo are the largest Native American tribe in the United States and the Navajo Nation is the largest Native American Reservation [1,2]. The boundaries of the Navajo Nation comprise portions of Utah, Arizona, and New Mexico, with a 2019 estimated population of 172,813 (+/−2643) within its borders [3]. Despite significant risk factors for ocular disease including high rates of diabetes, arid living conditions, and relative isolation from ophthalmology or optometry clinics, very few studies have been conducted on the status of ocular disease and ocular health among the Navajo and no large-scale study has ever been undergone to measure refractive error or visual acuity in this population.

Since 2013, the University of Utah John A. Moran Eye Center and Utah Navajo Health Services partnership has provided eye care to the citizens of the Navajo Nation via systematic mobile outreaches. Over the course of this collaboration, the University of Utah has conducted over 3300 clinic visits and provided treatment to thousands of Navajo patients. The services provided and layout of these outreach clinics are summarized in Figure 1.

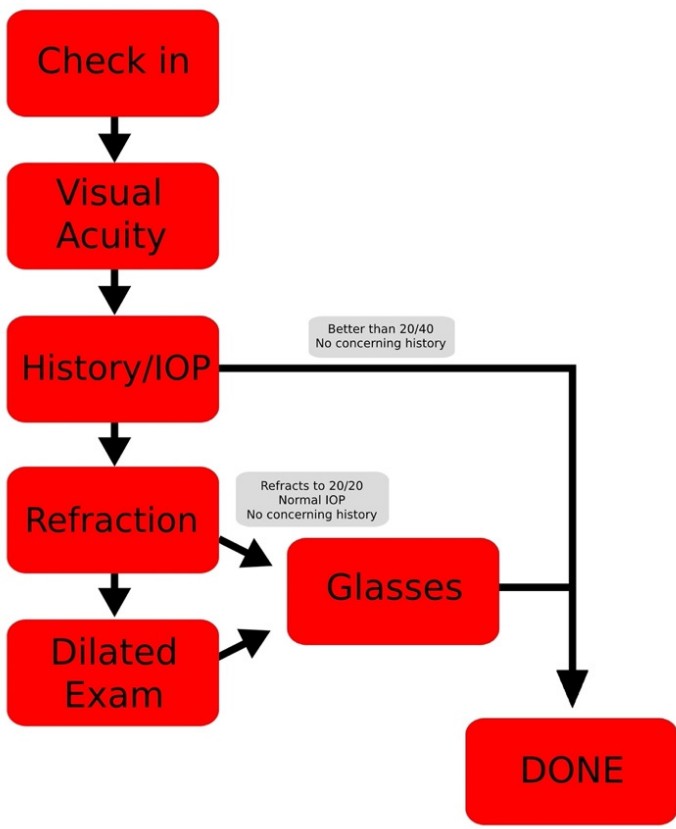

**Figure 1.** Outreach clinic layout and services.

## 2. Materials and Methods

IRB permission was obtained from both the University of Utah IRB and the Navajo Nation Human Research Review Board to conduct a full retrospective study of all first visits by adult (>18 years old) patients at Navajo outreach clinics from the years 2013–2021.

A chart search was conducted for the variables of gender, age at presentation, clinic location, visual acuity, intraocular pressure (IOP), and manifest refraction. After the data were collected, spherical equivalents (a method that accounts for the degree of nearsightedness, farsightedness, and astigmatism by a single number) were calculated from manifest refraction values and vision in the best-seeing eye was categorized according to the World Health Organization definitions of mild visual impairment (VI) (worse than 20/40 to 20/60 included), moderate VI (worse than 20/60 to 20/200 included), severe VI (worse than 20/200 to 20/400 included), and blind (worse than 20/400). All charts of patients meeting the WHO definition of blindness were further examined for the diagnosed cause in the best-seeing eye.

There are also several characteristics of these clinics that are important to note. In particular, to facilitate targeted screening within these clinics, patients are divided into those that present with glasses and those that present without (the distinction being used as a blunt heuristic to determine which patients are the least likely to have received prior eye care). This division was kept consistent in the chart review and the numbers were reported separately.

Furthermore, all manifest refraction in these clinics is conducted prior to patient dilation for the full exam (accommodation free). Full dilated exams were conducted via slit lamp examination and indirect ophthalmoscopy. Cases suspicious for macular edema or retinal pathology were further confirmed via optical coherence tomography (OCT) prior to conclusion of patient visits.

Starting in 2016, remote electronic medical record access was established at all ophthalmic outreaches to the Navajo Nation and visits were conducted with standard format-

ting and diagnoses. Thus, an additional review of all 2016–2021 adult patient charts was conducted for the variables of ocular history and final visit diagnoses made via dilated exam. The rates of nearsightedness (myopia) and farsightedness (hyperopia) were calculated using spherical equivalents (SE) from manifest refraction data and were added to the total diagnoses count. Consistent with several past epidemiologic studies, the definition of myopia was SE $\leq$ −0.50 diopters, and a diagnosis of hyperopia was made using a definition of SE $\geq$ +0.50 diopters [4–6].

## 3. Results

University of Utah Navajo Nation outreach clinics saw an estimated 2251 first-time patients during the years 2013 to 2021. The average age was 51 years old with a median of 53 (range, 18 to 92). The Montezuma Creek Clinic location saw 1303 patients, Monument Valley 667, and Navajo Mountain 281. These clinics had a female:male predominance of 1387:864.

Patient history characteristics are summarized in Table 1. Of note, 23.52% (271/1152) of patients presented with a history of type 1 or type 2 diabetes, 17.19% (198/1152) presented with a history of hypertension, and 4.95% (57/1152) of patients presented with a history of eye trauma.

**Table 1.** Histories and comorbidities, 2016–2021.

| Presenting Characteristics | *n* = 1152 |
|---|---|
| Diabetes (total) | 271 (23.52%) |
| Type I | 8 |
| Type II | 263 |
| Hypertension | 198 (17.19) |
| Trauma | 57 (4.59) |
| Hyperlipidemia | 40 (3.47) |
| Diabetes | 271 (23.52) |

The results of visual impairment among patients with and without glasses are found in Table 2. To summarize, 20.67% (198/958), 9.71% (93/958), and 3.13% (30/958) of patients presenting without glasses were classified with mild VI, moderate to severe VI, and blindness, respectively. In addition, 15.62% (202/1293), 5.18% (67/1293), and 0.46% (6/1293) of patients presenting with glasses were classified with mild VI, moderate to severe VI, and blindness, respectively.

**Table 2.** Visual impairment, 2013–2021.

| Category | Without Correction (*n* = 958) |
|---|---|
| Mild | 198 (20.67%) |
| Moderate–Severe | 93 (9.71) |
| Blind | 30 (3.13) |
| **Category** | **With Correction (*n* = 1293)** |
| Mild | 202 (15.62) |
| Moderate–Severe | 67 (5.18) |
| Blind | 6 (0.46) |

From 2013 to 2021, 1181 right eyes and 1182 left eyes underwent manifest refraction (glasses prescription) measurement. The average sphere in right eyes was −1.81, and the average cylinder, (a measure of astigmatism), was +1.54. The average sphere in the left eye

was −1.80, with an average cylinder of +1.52. The average right eye spherical equivalent was −1.19 and the average left eye spherical equivalent was −1.17.

A total of 1475 right eyes and 1461 left eyes underwent screening eye pressure checks, of which 2669 were by Tono-Pen, 232 by iCare, 32 by applanation tonometry, and 4 by other/unspecified methods. The average pressure in right eyes was 15.32 with a max of 60 and the average pressure in left eyes was 15.20 with a max of 66.

The summary of diagnoses from 2016 to 4/2021 is provided in Table 3. Of note, cataracts were the most common diagnosis, affecting 39.50% (455/1160) of patients and nearsightedness was the second most common diagnosis, comprising 22.75% (262/1152) of patients. Of those diagnosed with cataracts, 31.65% ($n = 144/455$) had a history of diabetes and 77.41% ($n = 352/455$) were over the age of 55. In addition, 17.71% (48/271) of patients with a history of diabetes were diagnosed with diabetic retinopathy (DR). Of these patients with DR, 20.83% (10/48) presented with diabetic macular edema (DME).

**Table 3.** Diagnoses, 2016–2021.

| Diagnosis | $n = 1155$ |
|---|---|
| Cataracts | 455 (39.50%) |
| Unilateral | 86 |
| Bilateral | 369 |
| Myopia (SE $\leq$ −0.50) | 262 (22.75) |
| Hyperopia (SE $\geq$ +0.50) | 107 (9.29) |
| Retinopathy | 48 (17.71, of diabetics) |
| Proliferative Diabetic Retinopathy | 15 (31.25, of pts with retinopathy) |
| Non-Proliferative Diabetic Retinopathy | 31 |
| Cystoid Macular Edema | 10 (20.83, of pts with retinopathy) |
| Pterygium | 146 (12.67) |
| Pinguecula | 124 (10.76) |
| Glaucoma | 50 (4.34) |
| Retinal Detachment | 17 (1.48) |
| Posterior Vitreous Detachment | 48 (4.17) |
| Epiretinal Membrane | 23 (2.00) |
| Age-Related Macular Degeneration | 5 (0.43) |
| Retinal Vein Occlusion | 2 (0.17) |

As summarized in Table 4, among uncorrected patients, cataracts were the most common cause of blindness (40%, 12/30) and the need for glasses was the second most common cause (33%, 10/30). Although retinitis pigmentosa was diagnosed in only 0.61% (7/1152) of patients, it represented the third most common cause of blindness in uncorrected patients (10%, 3/30). Among patients with glasses, glaucoma was the most common cause of blindness (50%, 3/6) and cataracts were the second most common (33%, 2/6).

**Table 4.** Diagnoses, 2016–2021.

| Cause | Without Correction (*n* = 30) | With Correction (*n* = 6) |
|---|---|---|
| Cataract | 12 (40%) | 2 (33) |
| Refractive | 10 (33) | |
| Retinitis Pigmentosa | 3 (10) | |
| Glaucoma | 1 (3) | 3 (50) |
| Diabetic Retinopathy | 1 (3) | |
| AMD | 1 (3) | 1 (17) |

## 4. Discussion

The advances and prioritization of health equity have recently paved the way for examination of health topics that are less frequently studied, including Native American health. The Navajo Nation represents the largest Native American tribe within the United States by both population and land area. As such, observations on the ocular health in this population comments not just on the reach of eye care in rural areas, but also on the status of Native American ocular health more generally.

Unfortunately, past endeavors to study Navajo ophthalmic health have had significant limitations. In the case of Friederich's study in 1982, analysis was limited to a single site experience with a small sample size [7]. In this study, Friederich concluded that trauma was the most likely cause of loss of vision in Navajo patients. However, Friederich also notes that the Navajo patients in his clinic were more likely to present for emergent care, an observation that, if true, would increase bias toward the study's conclusion. Another study in 1997 by Rearwin et al. sought to update Friederich's study by examining Indian Health Service (IHS) data. Like Friederich, this study concluded that trauma was the most common cause of blindness in the Navajo. However, the IHS data used by this study also relied heavily on emergent care visits and completely excluded routine causes of blindness, including the need for glasses, retinal disease, cataracts, etc. [8].

The methodology of this study resolves some of the issues of the two prior studies. Compared to a single site experience which may be more prone to regional bias, our data were provided via mobile outreaches across multiple clinic locations. In addition, compared to the lack of routine eye measurements in past studies, our data include visual impairment, eye pressure, manifest refraction, and dilated exam diagnoses. Furthermore, as the clinics provided services over time in a staggered yet consistent approach, the data in this study were more likely to capture routine causes of blindness, decreasing this study's bias toward emergencies alone.

Notably, the methodology in the secondary classification of the data in this study differs significantly from past studies. Unlike reporting rates of monocular vision loss, which disregard the overall ability of the patient to function with their remaining, unaffected eye, this study uses the WHO definitions of visual impairment and blindness in the best-seeing eye. In using these classifications, this study is less likely to overreport monocular issues and more likely to capture the most important factor: a patient's ability to function with the best-seeing eye.

As a result, this study provides many details regarding Navajo eye health that were previously completely unknown. First, the results indicate that the Navajo Nation outreach patients without glasses experienced higher proportions of mild VI, moderate–severe VI, and blindness than the most recent WHO-cited, Bourne et al. 2020 estimates. Even clinic patients that presented with some form of correction had higher rates of mild and moderate–severe visual impairment compared to these estimates [9]. If compared directly, these results suggest that the rate of blindness in patients without correction is almost six times worse than that of the 2020 Bourne et al. worldwide averages and that even among patients with glasses, many are highly undercorrected. Although the Bourne et al. estimates come from data sources that are predominantly survey-based and population

estimates, preventing statistical comparisons between this study and their estimates, these results may still represent a staggering contrast of vision outcomes between that of the Navajo Nation and the averages in the rest of the world.

Regarding treatment possibilities, the most common causes of blindness in Navajo ophthalmology patients without correction were cataracts and a need for glasses. These results suggest that a combined 73% (22/30) of blindness in patients without glasses is treatable and reversible via access to glasses prescriptions and surgery. Further, among all 2016–2021 patients, cataracts of any form were the most common diagnosis and near-sightedness the second most common. Both these conditions are easily treatable to prevent functional blindness or increased vision loss.

These numbers also provide the first look into the possible prevalence of myopia in this population, with 22.75% (262/1155) of patients being diagnosed. If compared directly, this prevalence is higher than Hashemi et al.'s estimated prevalence in the Americas of 16.2% (95% CI: 15.6–16.8), but lower than the highest worldwide population estimate of 32.9% (95% CI: 25.1–40.7) in South-East Asia [10]. To what degree the difference in these numbers is due to variation in genetics, early outdoor exposure in youth, or other factors will need to be examined by further studies and is, unfortunately outside the scope of this study's methodology.

This study also provides the first look at ocular comorbidities and diagnoses from dilated exams conducted by ophthalmologists. Consistent with past studies that predict a high rate of diabetes in this population, diabetes was the most common comorbidity in patient histories. However, this study also provides dilated exam data to measure the rates of eye complications in these patients and shows that a large proportion of diabetic patients have developed diabetic retinopathy and DME. Furthermore, a history of trauma was only found in less than 6% of patient histories, contradicting previous estimates [7,8]. In addition, this review provides clear data regarding the amount of refractive error among the Navajo, showing the average Navajo patient to be nearsighted with some astigmatism. The average degree of astigmatism was +1.52 in right eyes and +1.54 in left eyes, with 15.59% of patients having at least one eye with >2 diopters of astigmatism (351/2251). This percentage is much higher than the 4% predicted by Garber and Hughes' study of 175 Navajo patients in 1981 [11]. This study also represents the first eye pressure data published regarding Navajo eye health and shows that the average IOP for both eyes was within normal ranges. Although primary open angle glaucoma (POAG) affected 4.34% (50/1152) of patients from 2016 to 2021, only one case of blindness due to glaucoma was identified among uncorrected patients (3%, 1/30). However, POAG was the most common cause of blindness in patients who came to the clinic with glasses, affecting 50% (3/6) of these patients.

Although these results provide the first comprehensive look into Navajo ocular health epidemiology, perhaps most surprising were our findings regarding the relatively rare retina condition, retinitis pigmentosa, in Navajo clinic patients. Interestingly, although retinitis pigmentosa represented only seven diagnoses, its prevalence by 2021 of 0.61% (7/1152) in this clinic population is much higher than the estimated prevalence of 1/4000 for the United States population, a finding consistent with Friederich et al. and Heckenlively et al. [7,12,13]. In addition, although this disease did not represent a common diagnosis for patients at outreach clinics, it was the third most common cause of blindness in patients without correction (10%, 3/30).

As retinitis pigmentosa represents the most common inherited degenerative retinopathy, it was thought that these patients might be closely related. However, when the charts of these seven patients were analyzed for familial ties to one another, only one connection was discovered (two siblings) and any connections between the remaining patients is unknown from the data collected alone. Possible explanations for this high rate of RP may include self-selection bias (patients that see poorly such as RP patients will self-select to present to our clinics), familial ties (families often come together to the clinic and RP's familial inheritance is well-known), or a true phenomenon of higher rates of RP in this population

consistent with some limited studies [11]. Further studies will have to be conducted to truly estimate the rate of this disease in Navajo Native Americans and its impact on the Navajo Nation.

We recognize that this study has several limitations that prevent full generalization to the entire Navajo Nation population. As discussed above, although limited in their methods, past studies have asserted that trauma is the highest cause of vision loss and blindness in male Navajo patients [7,8]. If this assertion regarding gender of patients and trauma is correct, the female predominance in this study's data may increase bias toward a lower rate of trauma. This may be illustrated by the fact that among cases in these clinics, only seven patients presented with an immediate history of trauma and five required urgent follow-up. In addition, our high rates of blindness compared to the Bourne et al. numbers may result from self-selection bias. Since patients who are seeing the worst are the most likely to present to the clinic, this would increase the study's bias toward higher rates of blindness. Lastly, our system of patient-provided histories, without access to primary care patient records, including A1C numbers, may increase this study's bias toward incorrect rates of comorbidities such as diabetes, hypertension, etc.

Nevertheless, despite these biases preventing complete generalizability of this study's findings, several advantages remain. First, despite the reliance on patient histories, the percentage of diabetic and hypertensive patients in this data is in accordance with previous Navajo patient screening studies, suggesting that this study's comorbidity numbers may be generalizable [14–16]. In addition, by separating patients that present to the clinic with glasses from those without, using the WHO definitions to classify patients without glasses would likely lead to the inclusion of many patients that require no glasses at all. The inclusion of these patients would decrease the bias toward the high rates of blindness and visual impairment noted by this study. Despite these advantages, further studies, perhaps with population survey methods such as the Bourne et al. worldwide estimates, will likely have to be undertaken to provide findings that are most accurately generalizable to the entire Navajo population.

## 5. Conclusions

Navajo Nation outreach patients without correction present with higher proportions of visual impairment and blindness than the WHO-cited 2020 worldwide estimates results. Cataracts and need for glasses were the most common causes of blindness in this same group. Cataracts were the most common diagnosis via dilated full examinations and diabetes was the most common comorbidity. Trauma represents less than 6% of comorbidities, revising previous estimates. These results indicate that reliable data can be gathered through longitudinal outreach among Native American populations such as the Navajo Nation. We encourage academic and NGO outreach groups serving the underreported Native American populations to begin or continue to gather and publish patient data. Through these means and further work, much can be done to improve the ocular health outcomes in the Navajo Nation and other Native American populations and provide true health equity to these patients.

**Author Contributions:** Conceptualization, R.T.W., M.M., L.M., E.P., J.H.P., and C.J.C.; methodology, R.T.W., M.M., L.M., E.P., J.H.P., and C.J.C.; formal analysis, R.T.W.; data curation, R.T.W., M.M., and E.P.; writing—original draft preparation, R.T.W.; writing—review and editing, R.T.W., M.M., L.M., E.P., J.H.P., and C.J.C.; visualization, R.T.W.; supervision, J.H.P. and C.J.C.; project administration, R.T.W. All authors have read and agreed to the published version of the manuscript.

**Funding:** This research received no external funding.

**Institutional Review Board Statement:** The study was conducted in accordance with the Declaration of Helsinki and approved by the Institutional Review Board (or Ethics Committee) of the University of Utah and Navajo Nation Human Research Review Board. (IRB_00112719, 7/24/2018 and NNR-19.367, 1/21/2020).

**Informed Consent Statement:** Patient consent was waived by the University of Utah IRB and Navajo Nation Review Board due to the retrospective nature of this study.

**Data Availability Statement:** Patient data are protected under HIPAA and are unavailable for request. The deidentified data used in this study are property of the Navajo Nation. Any requests for deidentified data must be made to the Navajo Department of Epidemiology.

**Acknowledgments:** We acknowledge our valued partnership with the Utah Navajo Health Services and endorsement of the article by the Navajo Nation Human Research Review Board.

**Conflicts of Interest:** The authors declare no conflict of interest.

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
