# Peer review of "Patterns of Blindness in the Navajo Nation: A 9-Year Study"

_2411-5150, 2013_

Round 1
Reviewer 1 Report
Interesting article, well structurate.
About results: patients divided in uncorrected or corrected visual acuity; what do you sey about urgent or non urgent patology; how many of those with cataracts have associated diabetes , old ages;
conclusions is suported by the results.
Reviewer 2 Report
The authors investigated patients who received ophthalmological exam in their facilities for Navajo people. The results have value to know the trend of visual impairment of Navajo people, however, some revisions are needed, I think.
More detail information on the examination method would be needed. How was the refractive error measured? Were they measured without dilatation of the pupil or under the state of accommodation free? What was a method to observe fundus? Did the authors observe fundus by indirect ophthalmoscopy? Or was the judgement done on fundus photos? How did they diagnose macular edema? Did they use OCT?
Please make clear whether there were any fundus disease such as retinal vein occlusion and age-related macular degeneration other than diabetic retinopathy.
I prefer discussion on the prevalence of myopia of this cohort and world-wide data. I guess the prevalence of myopia may be lower than Asians. I understand the prevalence could not be obtained from the design of this study, still please add some comments on the frequency of myopia compared to other races. Can you estimate that the changes in life style in Navajo induce the prevalence of myopia?
Minor criticisms;
1. Line 58. Please spell out “EMR”.
2. Table 1 needs footnote to explain HTN. Or please revise to “Hypertension”.
Reviewer 3 Report
This manuscript by Wallace et al., summarized a 9-year (2013-2021) study of the patterns of blindness in the Navajo Nation, which is the largest American reservation in the US. The data are clear, novel and informative. It is suitable for publication in this journal after minor revision. I only have several minor questions:
1. According to the instructions of MDPI journals “ Acronyms/Abbreviations/Initialisms should be defined the first time they appear in each of three sections: the abstract; the main text; the first figure or table”. There are several abbreviations need to be defined, such as HTN in Table 1, PDR, NPDR and CME in Table 3 and AMD in table 4, even if most readers may know these abbreviations.
2. In table 1, it appears that there are two “Diabetes”, please verify it.
3. Now the genetic tests are very useful, I was wondering whether some of these patients may carry some gene mutations that associated with their blindness? Have the genetic tests been used in these patients of the Navajo Nation?
Thanks for the invitation.
Round 2
Reviewer 2 Report
My points are addressed.